# *APOE*: The New Frontier in the Development of a Therapeutic Target towards Precision Medicine in Late-Onset Alzheimer’s

**DOI:** 10.3390/ijms22031244

**Published:** 2021-01-27

**Authors:** Anna Yang, Boris Kantor, Ornit Chiba-Falek

**Affiliations:** 1Division of Translational Brain Sciences, Department of Neurology, Duke University Medical Center, Durham, NC 27710, USA; anna.yang569@duke.edu; 2Department of Neurobiology, Duke University Medical Center, Durham, NC 27710, USA; boris.kantor@duke.edu; 3Viral Vector Core, Duke University Medical Center, Durham, NC 27710, USA; 4Duke Center for Advanced Genomic Technologies, Durham, NC 27708, USA; 5Center for Genomic and Computational Biology, Duke University Medical Center, Durham, NC 27708, USA

**Keywords:** APOE, Alzheimer’s disease, late-onset, gene therapies, antisense oligonucleotides, monoclonal antibodies, gene editing, base editing, beta-amyloid, neurodegenerative disease

## Abstract

Alzheimer’s disease (AD) has a critical unmet medical need. The consensus around the amyloid cascade hypothesis has been guiding pre-clinical and clinical research to focus mainly on targeting beta-amyloid for treating AD. Nevertheless, the vast majority of the clinical trials have repeatedly failed, prompting the urgent need to refocus on other targets and shifting the paradigm of AD drug development towards precision medicine. One such emerging target is apolipoprotein E (*APOE*), identified nearly 30 years ago as one of the strongest and most reproduceable genetic risk factor for late-onset Alzheimer’s disease (LOAD). An exploration of *APOE* as a new therapeutic culprit has produced some very encouraging results, proving that the protein holds promise in the context of LOAD therapies. Here, we review the strategies to target *APOE* based on state-of-the-art technologies such as antisense oligonucleotides, monoclonal antibodies, and gene/base editing. We discuss the potential of these initiatives in advancing the development of novel precision medicine therapies to LOAD.

## 1. Introduction

Alzheimer’s disease (AD) is the most common age-related neurodegenerative disease (NDD). Age is the leading risk factor for AD; thus, with a rapidly growing aging population, the number of AD cases is growing fast and projected to rise drastically over the next three decades. In the US alone more than 5 million people are currently living with AD and this number is projected to reach 14 million cases by 2050. As a consequence, AD poses a huge economic burden on society, placing overwhelming strain on the healthcare system. In 2020, the cost of AD to the US was $301 billion, including $206 billion in Medicare and Medicaid payments, and caregivers provided $244 billion worth of care [1]. These trends will worsen because there are no therapies to halt or prevent AD, projected to cost more than $1.1 trillion annually by 2050. Despite all the research effort, money, and commitment, there is no cure for AD, nor any disease-modifying therapies (DMT) to slow down or even delay the progression of the disease. The current treatments are mostly palliative aimed at improving symptoms management rather than curing the disease. To date, there are only five FDA-approved drugs available for AD, three of which are cholinesterase inhibitors (donepezil, rivastigmine, and galantamine) [2], the fourth being an N-methyl-D-aspartate (NMDA) receptor antagonist [3], and the fifth a combination of cholinesterase inhibitor and NMDA-receptor antagonist.

As mentioned above, these drugs may only provide some level of symptomatic relief for the patients, and numerous clinical trials to identify disease-modifying therapies (DMT) for AD have failed. Thus, AD remains an unmet medical need. Nonetheless the mainstream research for AD therapies continues to be based on the amyloid cascade hypothesis and, as such, the leading molecular target has been for decades the beta-amyloid isoforms (Aβ), the cleavage product of the amyloid precursor protein (APP) [4] (Figure 1). These clinical trials are aimed at decreasing the levels of Aβ aggregates and plaques using drugs that reduce production and aggregation and/or increase clearance of Aβ. Unfortunately, such trials have repeatedly failed, underscoring the lack of mechanistic understanding of AD pathogenesis and the urgent need for a paradigm shift in AD clinical research. While these failures have driven researchers to initiate clinical trials earlier in the course of the disease, with the concept that earlier intervention might be more effective, another major reason for the failure to identify an effective treatment is likely the inaccurate consideration of AD as a homogeneous disease. In this respect, increasing evidence demonstrates the heterogeneity in the underlying pathophysiologic processes of AD and show variability in the genetic risk and molecular profiles amongst AD patients [5,6].

Collectively, advancement in AD therapy requires the development and validation of new therapeutic targets, including drug targets tailored to sub-group/s of patients with specific risk factors. Thus, to date many investigators and funding bodies recognize the need to shift the focus to potential culprits other than Aβ. Consistently, recently, alternative targets, such as *APOE*, have emerged as potential promising targets for AD treatment [8,9,10]. Nonetheless, the development of *APOE* as a new drug target moves at a slower pace compared to the continuing clinical research related to Aβ. Of the 46 current pharmacological NIA-funded active Alzheimer’s and related dementias clinical trials and studies, 46% of the studies continue to focus on Aβ as their target, with tau and neuronal loss as targets following far behind with respective percentages of 13% and 15%, as seen in Figure 1. The remaining targets altogether account for less than 26% of current clinical trials, with currently no active clinical trials targeting *APOE* [7]. This review is focused on *APOE*, the most established risk factor for late-onset AD (LOAD) [11], as a potential therapeutic target for LOAD towards precision medicine in AD. We describe the evidence supporting *APOE* as a promising target and, in particular, the new research initiatives that have been developed to target the *APOE* gene and protein product, namely, antisense oligonucleotide (ASO), monoclonal antibody (mAbs), and gene editing approaches.

## 2. The *APOE* Locus Is the Strongest Genetic Risk for LOAD

### 2.1. ApoE Protein: Function and Isoforms

The apolipoprotein E protein (ApoE) has multiple functions and plays key roles in lipid metabolism, neurobiology, and neurodegenerative diseases. Its major function is to transport lipids among various cells and tissues of the body. In addition, intracellular ApoE may modulate various cellular processes physiologically or pathophysiologically, including cytoskeletal assembly and stability, mitochondrial integrity and function, and dendritic morphology and function [12]. Overall, ApoE is widely involved in human health and disease.

ApoE is encoded by the *APOE* gene positioned on chromosome 19q13.32 (GRCh 38: chr19:44,905,795–44,909,392). Two common coding SNPs in exon 4 of the gene give rise to three allelic variants, *APOE* e2*, APOE* e3*,* and *APOE* e4, encoding three corresponding protein isoforms that differ at two amino acid positions, 112 and 158: ApoE2 (Cys^112^; Cys^158^), ApoE3 (Cys^112^; Arg^158^), and ApoE4 (Arg^112^; Arg^158^). It was suggested that the single amino acid change (position 112) between the ApoE3 to ApoE4 protein isoforms resulted in structural differences that involve the interactions between sequences from both the N- and C-terminal domains [13,14] (Figure 2A).

### 2.2. APOE e4

The first and most firmly established genetic risk factor for LOAD is the e4 allele of the apolipoprotein E gene (APOE e4) [15,16,17,18]. The initial discovery was made nearly 30 years ago by linkage analysis of pedigrees [15] and over the ensuing years it has become the most highly replicated genetic risk factor [15,16,17,18]. Carrying the APOE e4 variant significantly increases the lifetime risk for LOAD, whereas the number of e4 copies affects the level of risk and is associated with the age of clinical disease onset [15,19], while APOE e2 conferred a protective effect [18,19,20]. Although the precise molecular mechanisms underlying ApoE e4-mediated risk effects have not been fully elucidated, it was suggested that ApoE e4 acquired hyperfunction (gain of toxic effects) [21] and increasing data suggested several cellular pathways through which ApoE e4 may exert toxicity associated with LOAD pathologic phenotypes [22,23,24,25,26,27,28]. Collectively, these studies provide strong support to the concept that decreasing the levels of ApoE e4 specifically will have a therapeutic implication. However, ApoE e4 as a target for LOAD still remains significantly understudied, despite the few recent studies that have begun to pave the way.

### 2.3. Dysregulation of APOE Expression

Over the last decade, LOAD genome-wide association studies (GWAS) have confirmed strong associations with the APOE LD genomic region, and no other LOAD-association remotely approached the same level of significance [29,30,31,32,33,34,35,36,37]. Although some LOAD GWAS exclude all variants in this region, because of their high LD with the coding SNPs that define the APOE genotype (e2, 3, 4), other LOAD genetic studies have focused on the association between variants and haplotypes based on the promoter and enhancer regions of genes in this region and LOAD phenotypes [38,39,40,41,42,43,44,45]. However, whether the strongest signal is attributed to additional variants and haplotypes within this LD region jointly with e4, as well as the molecular mechanisms underlying the LOAD-association with the APOE LD region, is largely unknown.

Accumulating new evidence has suggested that the increased overall expression of APOE plays an important role in the etiology of LOAD (reviewed in [21]). Foremost, previously we found significant higher levels of APOE-mRNA in brain tissues obtained from e3/3 LOAD patients compared to 3/3 healthy donors, consistently with other reports showing elevated levels of APOE-mRNA in LOAD brains [46,47,48,49]. In addition, new single-nucleus (sn)RNA-seq datasets showed LOAD changes in APOE expression in glia cell-types, in particular upregulation in microglial subpopulations [50,51,52]. Moreover, studies using the APP/PS1 transgenic mice showed that lowering the ApoE protein levels ameliorated cognitive dysfunctions and Aβ pathology [53] independent of the APOE allele [8,54,55]. Lastly, studies showed LOAD associated differential DNA-methylation [56,57,58,59,60], further supporting that dysregulation of APOE expression plays a role in the genetic etiology of LOAD. In conclusion, while ApoE4 has received much attention for its LOAD-risk effect, there are clear changes in APOE expression associated with LOAD and independent of the e4 allele [8], suggesting that regulation of APOE expression may impact the risk to develop LOAD, making the modulation of the overall ApoE protein levels useful as a future therapeutic target.

Based on the evidence presented above, we postulate that the e4 allele’s inherent hyperfunction, by which it exerts its pathogenic effect, is comparable to the plausible pathogenic effect of elevated e3 expression. Thus, increased activity of ApoE, mediated by either a coding mutation in exon 4 and/or gene dysregulation, is the key role of APOE in the genetic etiology of LOAD (Figure 2A). Thus, methods for modifying the e4 isoform and/or reducing APOE levels introduce a new promising avenue towards precision medicine in LOAD (Figure 2B).

Noteworthy, APOE e2 has been identified as a longevity variant, associated with beneficial effects on cognition, and accumulating evidence suggested it protects against AD (reviewed in [61]). While the mechanisms driving its protective effect remain unclear, potential therapeutic strategies designed to leverage the protective effect of APOE e2, such as viral-mediated overexpression of APOE e2 and gene-editing conversion of APOE e4 to e2, hold promise as treatment options for AD. With that said, APOE e2 increases the risk of other diseases, including neurological disorders; thus, long-term safety concerns should be carefully evaluated when considering treatments inspired by the protective role of e2 in AD. This topic has been extensively reviewed elsewhere [61,62] and is outside the scope of this review.

In the current review we focus on therapeutics strategies designed to mitigate the risk of LOAD conferred by the APOE gene (as proposed in Figure 2A). Below we describe three major technologies aimed at targeting the APOE gene, transcript, or protein, which may serve as a proof-of-concept for prospective therapeutics approaches (Figure 2B).

## 3. Technologies for Targeting *APOE* as a Proof-of-Concept for LOAD Therapies

### 3.1. Antisense Oligonucleotide Therapy

#### 3.1.1. ASO Technology and Application in Disease Therapy

Antisense oligonucleotides (ASOs) are short, single-stranded deoxyribonucleotides, typically consisting of about 25 nucleotides. ASOs are synthetically designed to be complementary and capable of hybridizing to specific mRNA strands [63], such that they can modify the expression of mRNA using either a splice modulation or knockdown approach, as determined by the target and design chemistry [64]. The ASO approach aims at downregulation of the production of the disease-causing protein by recognizing the particular RNA molecule target from which the protein is translated (Figure 3). ASOs are a promising therapeutic technology because of their rapid and highly selective capabilities to target and destroy a specific RNA molecule based on its sequence. However, there are few shortcomings for ASO as a therapeutics approach; foremost, effective ASO delivery methods for the CNS remain a challenge, as ASOs lack the ability to penetrate the blood-born barrier (BBB) efficiently [65]. Another concern is the possibility of adverse side effects due to ASO-induced cellular toxicities and off-target effects in both sequence- and chemistry-dependent manners [65]. In addition, the potential robust knockdown mediated by ASOs can be deleterious, resulting in a deficiency in the normal physiological levels of the target protein that is needed to maintain normal biological processes and cellular function. For example, RNAi studies reported neurotoxicity associated with a robust reduction in the SNCA levels [66,67], and suggested the need to maintain normal physiological expression levels of the SNCA protein.

ASOs are currently being used as a possible successful treatment method for several diseases. For example, the ASO-based drugs Etepliersen and Golodirsen, used to treat Duchenne muscular dystrophy (DMD), were approved in the US in 2016 and 2019, respectively. DMD results from a mutation in the gene encoding the protein dystrophin, resulting in a shorter and prematurely truncated protein with a deletion of exon 49 and 50. Etepleirsen acts by using a morpholino ASO for exon skipping purposes, ultimately excluding exon 51 and leading to a restored reading frame and leaving a more functional dystrophin protein [64]. Similarly, Golodirsen also utilizes ASO to perform exon skipping by hybridizing with DMD pre-mRNA and skipping exon 53 leading to the production of a shorter, but functional dystrophin protein [68]. Another example is the drug Nusinersen, approved by the FDA in 2016 and by the European Commission Agency in 2017, for treating certain forms of spinal muscular atrophy (SMA) [64]. SMA, a neuromuscular disorder, is caused by loss of function mutations or deletions of the survival motor neuron 1 (SMN1) gene. Nusinersen acts to enhance the production of functional SMN protein encoded by the highly homologous gene *SMN2*. The *SMN2* gene naturally exhibits aberrant splicing of exon 7, resulting in a truncated and unstable protein. The ASO targets *SMN2* pre-mRNA to induce the correct splicing, leading to increased levels of the full length SMN protein. This ASO therapy demonstrated significant improvement in motor function of treated children when compared to sham procedures [69]. There are also several ASO therapies in early phase clinical trial testing. An example is the drug targeting HTT in Huntington’s disease, which use ASO to decrease the levels of the mutant huntingtin protein (mHTT) [70,71]. Another example is ASO developed to target superoxide dismutase 1 (*SOD1)* for the treatment of amyotrophic lateral sclerosis (ALS) [64]. Mutations in *SOD1* are responsible for ~20% of familial ALS, leading to the gain of toxic function. ASOs for *SOD1* decreased the protein levels and extended survival in rat models [72], and showed beneficial outcomes by reducing the SOD1 levels in a Phase 1 clinical trial [73].

#### 3.1.2. Current Progress in ASO Therapy Targeting APOE

ASO to reduce *APOE* expression was applied in a study using a mouse model of beta-amyloidosis, the APP/PS1-21 transgenic, homozygous for either the *APOE*e4 or *APOE*e3 alleles. The ASO treatment decreased the *APOE*-mRNA and protein levels in the mice brains by at least 50% compared to the controls. The investigators examined the effect of the ASO treatment on phenotypes related to Aβ pathology, characteristics of the APP/PS1-21 model [8]. The authors reported age-dependent effects on reversing the Aβ phenotypic perturbations. ASO treatment starting after birth led to a significant decrease in Aβ pathology when assessed at 4 months, while ASO treatment starting later at the onset of amyloid deposition (6 weeks) showed an increase in Aβ plaque size with no change in overall Aβ burden. On the other hand, both age groups demonstrated a reduction in plaque-associated neuritic dystrophy upon ASO treatment [8]. These overall results suggest that the *APOE* levels may play an important role in the earlier stages of Aβ plaque formation but has a more limited effect once the Aβ pathology has already begun. Thus, since AD pathology begins prior to clinical symptom onset, the treatment aims to reduce *APOE* such that ASO may be most effective in the earlier pre-symptomatic stages [8]. Noteworthy, the ASO effect on the Aβ pathology, mediated by the reduction in *APOE* levels, was independent from the allele type and was detected in both e3 and e4 mice [8]. The observation that lowering the overall *APOE* level has beneficial effects, regardless of the allelic isoform, is consistent with the hypothesis we presented in Figure 2. Collectively, this study provides the foundation for the development of an ASO treatment aimed at reducing *APOE* expression to combat LOAD.

### 3.2. Monoclonal Antibody Therapy

#### 3.2.1. Monoclonal Antibody (mAb) Approach and Application in Disease Therapy

Monoclonal antibody (mAb) therapies designed to target certain cells (e.g., cancer cells) or proteins. The mAbs were developed to target specific protein epitopes and resulted in the deactivation of the protein-of-interest via blocking the protein function (e.g., occupying the ligand-receptor binding site) and/or recruiting the immune system, which, in turn, activates protein degradation pathways, leading to decreased levels of the targeted protein or proteins complex (Figure 4). Most mAb therapy applications utilize direct protein delivery [74]. However, mAbs can be paired with virus-mediated gene transfer, e.g., using adeno-associated (AAV) or lentiviral (LV) vectors or a non-viral delivery route by RNA or DNA expression plasmids [75]. In the case of non-direct delivery (viruses, plasmids), antibody sequences will be inserted and administered to the host cell, and antibody production will occur [76] (Figure 4). Antibody therapy has several strengths, including a well-established protocol-of-delivery and route of administration, ability to cross the blood–brain barrier (BBB), interacts with the immune system, has a long half-life, and their ability to target precise locations, which ultimately makes the viral vectors an efficient delivery vehicle [77,78]. However, there are several shortcomings that characterize antibody-based therapies, including a high-level of immune response, low specificity, low stability and solubility [79], and usually requires repetitive injections in order to achieve therapeutic/correction levels [80].

Monoclonal antibody therapies have been used successfully to treat many immune-related diseases, such as cancer and inflammatory diseases [81]. Clinical trials are ongoing for the use of mAbs in processes such as transplant rejection, rheumatoid arthritis, and prevention of viral infection [75]. The use of mAbs in this respect has been studied since the 1990s and has seen much success, and is frequently used successfully in drugs such as infliximab [82].

mAb-based treatment has also been developed to target the proteins that form aggregates in neurodegenerative proteinopathies, including Parkinson’s disease (PD) and AD. In PD, α-synuclein protein undergoes a conformational change, resulting in the formation of Lewy bodies, the neuropathological hallmark of PD [83]. Several anti-α-synuclein antibodies are being tested in clinical trials for their potential to target Lewy bodies and ameliorate their effects on PD progression; for example, PRX002 using the humanized anti-α-synuclein monoclonal antibody 9E4 [84,85] and the anti-α-synuclein antibody BIIB052 [86,87]. Several anti-β-amyloid antibodies have also been studied in clinical trials. Solanezumab by Eli Lilly and Company showed a non-significant slowing of cognitive decline and no effect on β-amyloid and was discontinued after Phase 3 [88,89]. Another β-amyloid antibody currently in Phase 3 testing is Crenezumab [90,91].

The most recent example of anti-β-amyloid antibodies is seen in Aducanumab developed by Biogen. Aducanumab is a human-derived monoclonal antibody that targets β-amyloid aggregates [92] by binding preferentially to the β-amyloid plaques rather than monomers, such that the monomer form remains available for potential neuroprotective functions [93]. Aducanumab completed its Phase 3 parallel-group studies EMERGE and ENGAGE in 2015. However, the two studies were abandoned in 2019 following a futility analysis of an independent monitoring committee. A larger dataset analysis later found that, of the two cohorts, the EMERGE data did meet its primary endpoint [94], with those treated with the higher doses of Aducanumab showing a reduction in β-amyloid plaques and a 23% reduction in cognitive decline [94]. However, the FDA decided not to recommend Aducanumab for regulatory approval in November of 2020 stating that the Phase 3 trial did not provide enough evidence of the drug’s effectiveness in treating AD [95].

#### 3.2.2. Monoclonal Antibodies Targeting ApoE

Several groups have evaluated anti-apoE immunotherapy as a method to rescue the pathological and behavioral features of AD. A passive immunization with the HJ6.3 apoE monoclonal antibody, before plaque onset, resulted in a 60–80% reduction in Aβ accumulation in the cortex and hippocampus of an APP/PS1 mouse model [10]. The effects of treatment with the anti-ApoE antibody HJ6.3 on measures for learning and memory and on Aβ pathology were also evaluated in the mice after the onset of Aβ plaques deposition. This study showed that the anti-ApoE antibody HJ6.3 prevented the formation of new plaques, inhibited Aβ plaque growth, and improved brain function [96]. Moreover, administration of this antibody led to changes in microglia responses in areas around the Aβ plaques and showed that the anti-apoE antibodies bind to ApoE in plaques and activated microglia [10]. These results suggested that treatment with anti-ApoE may activate microglia-mediated clearance of the Aβ aggregates [9,10]. While this study did not detect any influence on brain ApoE and peripheral cholesterol metabolism, we cannot exclude the possibility that the mAb treatment reduced the ApoE activity that is essential for Aβ aggregation.

Another study examined ApoE antibody therapy utilizing an antibody that recognizes specifically the ApoE4 isoform, anti-ApoE4 9D11. Peripheral injection showed that the 9D11 monoclonal antibody inhibited the accumulation of Aβ in the hippocampus [97]. Furthermore, repeated 9D11 injections showed the formation of ApoE/IgG complexes that reversed the cognitive impairments compared to the control ApoE4 mice, indicating a potential reversal of cognitive impairment [97]. Another study applied the anti-human ApoE4 antibody (HAE-4), targeting the nonlipidated aggregated ApoE before plaque onset also found a reduction in Aβ pathology characteristic of the APPS1-21/human*APOE*4 mice [98].

Overall, these studies showed that anti-ApoE monoclonal antibodies have consistent effects on decreasing the Aβ plaque load and generally improves cognitive abilities, suggesting that anti-ApoE immunotherapy is a promising therapeutics approach for LOAD.

### 3.3. Gene Editing

#### 3.3.1. Applications of Gene Editing in Disease Therapy

The process of gene editing offers the ability to remove, revise, or replace a certain disease-causing gene mutation with a healthy copy of the gene at the DNA level and can provide durable and stable benefits by introducing precise changes to the DNA sequence directly to target cells [99]. The CRISPR-Cas9 system and its modifications (e.g., Cas variants such as deactivated Cas9) have been the mainstream in gene-editing research, including research aiming at the development of clinical applications [99] (Figure 5). While the breakthrough technology of genome editing brings profound therapeutics opportunities to treat, cure, and prevent genetic diseases, there are challenges that affect the translation into clinical applications. At present, the major hindering aspects of therapeutics genome editing include accuracy (off-target events), precision (undesired genomic sequence change), safety (e.g., immunogenicity), efficacy, efficient delivery systems, and extreme costs. The opportunities and challenges have been recently reviewed in detail elsewhere [100].

For example, targeted genomic deletion using CRISPR-Cas9 represents a promising a therapeutic approach for the treatment of Leber Congenital Amaurosis 10 (LCA10), a severe retinal dystrophy, in patients with an intronic mutation in *CEP*290 [101]. CRISPR-Cas technology has also been used in AD research to target the APP Swedish mutation. The mutation leads to increased β-secretase cleavage of the amyloid beta precursor protein and thus high levels of Aβ. Deleting the mutation in an allele-specific manner by the CRISPR-Cas system reduced the level of the secreted pathogenic Aβ, and demonstrated the effectiveness of gene editing as a therapy for familial AD caused by APP dominant mutations [102].

#### 3.3.2. Using CRISPR/Cas Technologies to Target APOE

Using the CRISPR-Cas gene-editing technology to target the *APOE* gene has a potential for ameliorating the pathogenic effects of the ApoE4 isoform, despite the current incomplete understanding of the underlying mechanisms that drive its effect. Applying CRISPR-Cas allows the precise editing of the e4 risk allele to the “natural” e3, as these different alleles differ by a single nucleotide rs429358, T→C. Few studies explore the effect of *APOE* gene editing using human induced pluripotent stem cell (hiPSC)-derived models. Characterization of hiPSC-derived neurons with the e4/4 genotype demonstrated higher levels of tau phosphorylation, increased production of Aβ, and GABAergic neuron degeneration. Converting e4/4 to e3/3 by CRISPR-Cas 9 gene editing rescued these phenotypes. The ability to ameliorate the various AD-related phenotypes suggested that gene editing is a promising therapeutic method for targeting *APOE* [103]. Another study extended the characterization analyses to different hiPSC-derived brain cell types. This study also utilized CRISPR/Cas9 to convert a hiPSC line from a healthy donor homozygous for the e3 allele into an isogenic hiPSC e4/4 line. The *APOE* e4 neurons exhibited elevated synaptic activity, increased synapse number, early endosomes, and elevated Aβ_42_ secretion relative to the isogenic *APOE* e3 neurons. *APOE* e4 astrocytes exhibited impaired clearance of extracellular Aβ and cholesterol accumulation, and *APOE*e4 microglia-like cells showed altered morphologies, inflammatory gene activation, and less efficient Aβ uptake compared to the corresponding *APOE* e3 cell types. Consistently, converting *APOE* e4 to *APOE* e3 was sufficient to attenuate multiple AD-related pathologies in the neurons, astrocytes, and organoids derived from a hiPSCs line obtained from a sporadic AD patient [104].

Base editing is another CRISPR-Cas9-based technology to correct the e4 coding SNP. This technology employs the deactivated Cas9 (dCas9) or nickase Cas9 (nCas9) fused with cytidine deaminase enzyme that allows the conversion of a C to T at single-base point mutations without the double-strand DNA backbone breaking, thereby increasing the editing efficiency and reducing additional unwanted insertions and deletions [105]. A study that used *APOE* to compare the technologies demonstrated a higher percentage of correction with the base editing approach compared to the conventional gene editing approach with less non-specific events [106]. A recent development in the CRISPR-Cas field, prime-editing [107], may offer a more precise and safe approach with much less undesired effects to correct the *APOE* e4 coding SNP.

In summary, utilization of CRISPR/Cas9-mediated gene editing approaches can efficiently change the *APOE* genotype and successfully reverse AD-related phenotypes and impact Aβ aggregation. Altogether, these studies provide a proof-of-concept for the therapeutic potential of this technology in precision medicine for LOAD patients carrying the e4 allele.

## 4. Conclusions

*APOE* is a well-established and long-standing genetic risk factor for LOAD; however, it has only recently emerged as a therapeutic target. We suggest that a shift in direction towards targeting *APOE* in treating LOAD could result in new and effective treatments. The ultimate goal of precision medicine is to enable clinicians to accurately and efficiently identify the most effective preventive or therapeutic intervention for a specific patient. The ability to precisely characterize the *APOE* genotypes and determine carriers of the e4 risk allele facilitates the identification of the patients’ group that suffers from LOAD due to *APOE*, and hence will be potentially responsive to a treatment regimen that targets *APOE*. Thus, *APOE* is a prominent example for a target moving forward towards precision medicine in LOAD. New technologies offer the opportunity to develop gene-specific and even isoform/allele-specific therapies, and by that enable the advancement of strategies for precision medicine. Here, we discussed applications of these technologies to target *APOE* and specifically strategies directed at the DNA, RNA, and protein levels of *APOE*. Noteworthy, these technologies, on the other hand, bear significant disadvantage, e.g., a low efficiency and specificity, low stability and solubility, adverse immunoreactivity, and inability to penetrate the blood-brain barrier (BBB) [65,79]. Thus, emerging innovative genomic technologies and delivery techniques may circumvent these limitations. Nonetheless, these approaches can be implemented to other known and perspective candidate gene-targets of LOAD, including dysregulated genes and rare mutations such as *TREM2* [108].

## Figures and Tables

**Figure 1 ijms-22-01244-f001:**
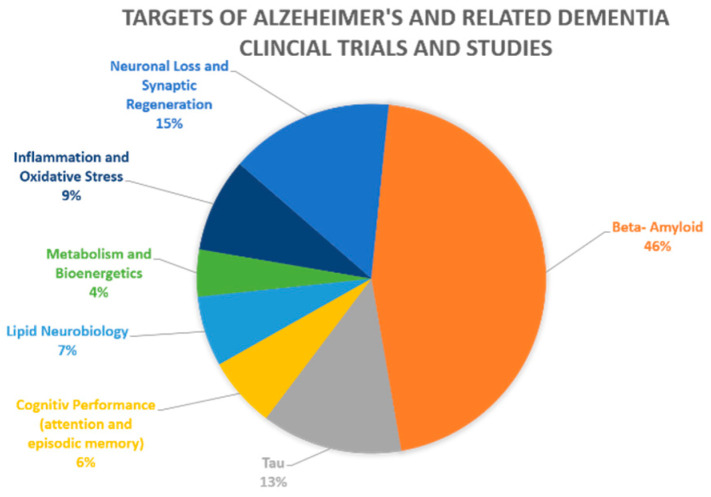
Clinical trials and studies for Alzheimer’s and related dementia by drug targets. The figure represents the 46 current pharmacological NIA-funded active Alzheimer’s and related dementias clinical trials and studies. The different molecular/pathways targets are color coded, with their percentage from all current clinical studies indicated. Beta-amyloid is the target of 46% of the studies, tau and neuronal loss follow far behind with respective percentages of 13% and 15% [7].

**Figure 2 ijms-22-01244-f002:**
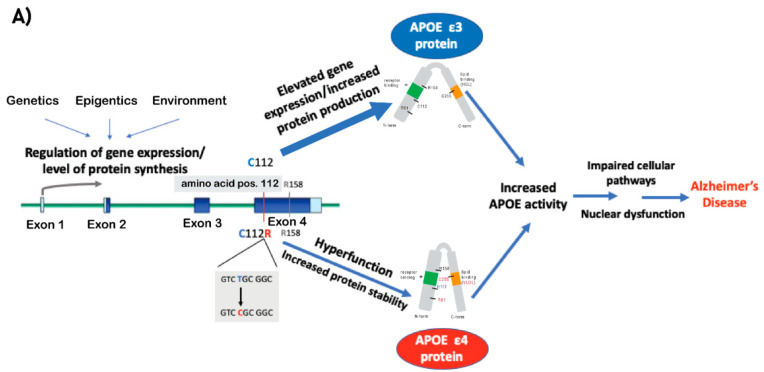
(**A**) A schematic model describing the mechanisms that lead to increased ApoE activity and by that mediate the pathogenic effect of *APOE* e4 and *APOE* e3 (differ in amino acid at position 112 Arg and Cys, respectively) on LOAD. Coding exons designated in blue boxes, UTRs in light blue, introns in green lines. (**B**) A diagram of the different technologies described in this review to target ApoE, including antisense oligonucleotide (ASO), monoclonal antibody (mAbs), and CRISPR/Cas9 gene editing technologies. The ultimate goal of the presented methods is to establish a proof-of-concept for the development of precision medicine treatments for LOAD by targeting ApoE and correcting its coding sequence or reducing its production. Purple boxes represent the therapeutic approaches, gray boxes represent how APOE mediates its the pathogenic effects (left) and the outcomes of the therapeutics approaches with respect to APOE (right).

**Figure 3 ijms-22-01244-f003:**
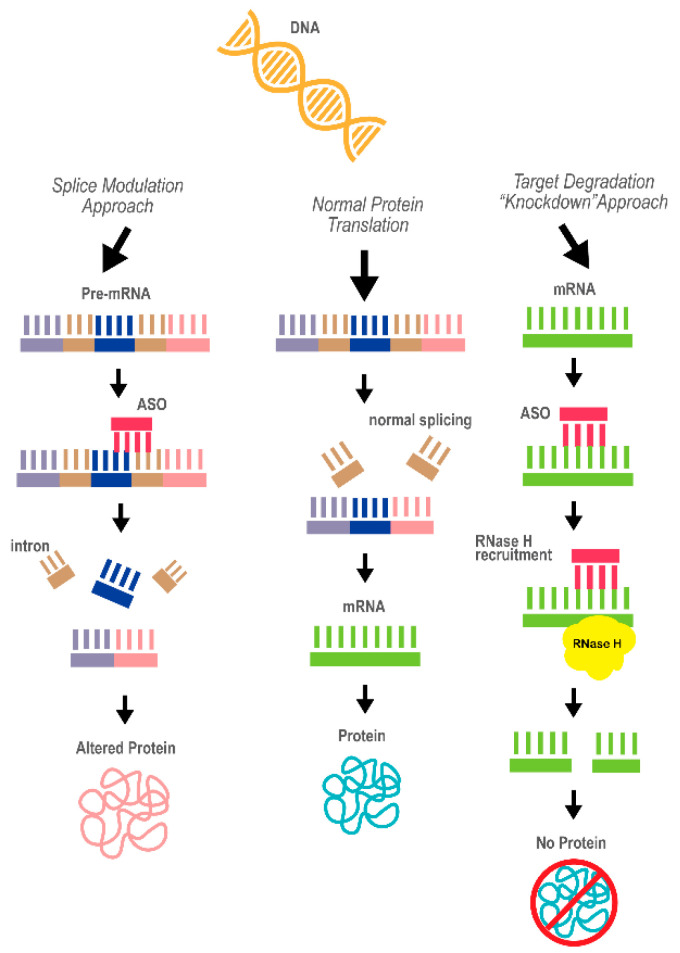
A diagram of the concept behind antisense oligonucleotide (ASO) technology as a therapeutic approach. The two mechanisms underlying the ASO effects, splice modulation and target degradation (“knockdown”), are depicted. The ASO is marked in red. Purple, dark blue and pink represent different exons of the pre-mRNA; brown represents introns; green represent the processed mRNA (the product of correct splicing)

**Figure 4 ijms-22-01244-f004:**
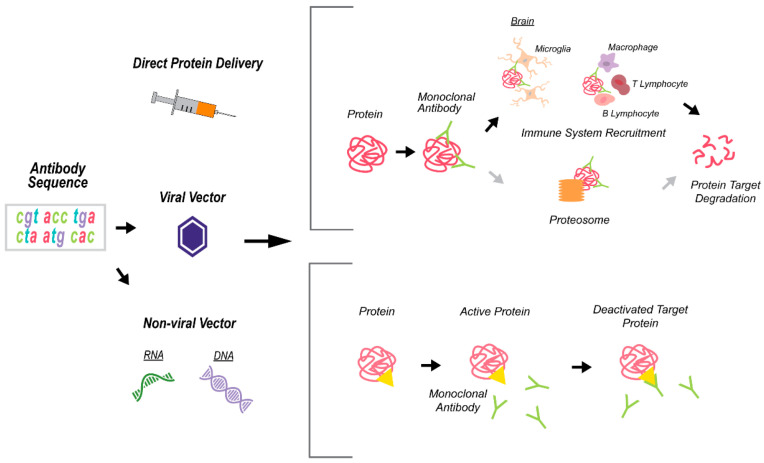
A schematic depicting the monoclonal antibody (mAb) technology for immunotherapies. Antibody delivery is done directly or through viral or non-viral vectors that contain the antibody sequence. The mAb specifically recognizes the target protein/s and exerts its effects via blocking the protein function (e.g., yellow triangle represents the ligand-receptor binding site) or by recruiting the immune system and activating protein degradation pathways.

**Figure 5 ijms-22-01244-f005:**
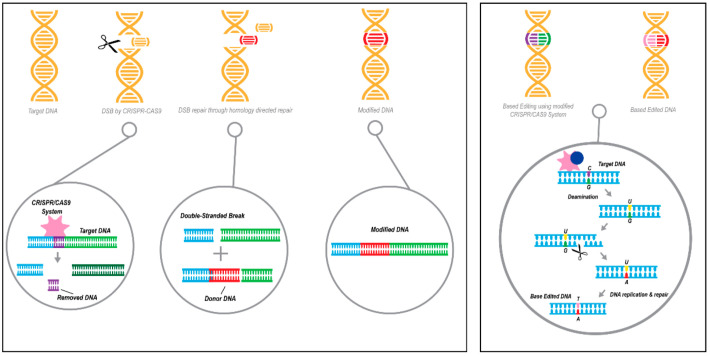
An illustration describing CRISPR/Cas9 technology as gene therapy approaches. Left panel: Gene editing acting on the DNA strand-level by knock-in replacement using the CRISPR-Cas9 system, resulting in targeted genomic deletion followed by homology-directed repair (HDR) with a donor DNA sequence. Orange represents DNA, red donor DNA,) Right panel: Acting on the chemistry of a single DNA base, gene editing by base editing technology utilizes the CRISPR/dCas9 or CRISPR/nickase(n)Cas9 system. Orange represents DNA, the following colors represent the 4 deoxyribonucleic acids: purple C, green G, pink T, red A.

## Data Availability

Not Applicable.

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
