# Peer review of "APOE: The New Frontier in the Development of a Therapeutic Target towards Precision Medicine in Late-Onset Alzheimer’s"

_ijms, 2021, doi:10.3390/ijms22031244_

Round 1
Reviewer 1 Report
I congratulate the authors for a paper that is very well written, interesting and well structured. I have a few, comments:
- Page 2, first paragraph, line 8. The authors state that: “there are five approved drugs available for AD, four of which are cholinesterase inhibitors” and “ the fifth being an N-methyl-D-aspartate (NMDA) receptor antagonist.”
To my knowledge, there are only 3 cholinesterase inhibitors approved by the FDA in the US (donepezil, rivastigmine and galantamine). In addition, 1 drug, combining memantine (NMDA-receptor antagonist) and donepezil, is available under the Brand Namazaric®. Also memantine (NMDA-receptor antagonist) in a single brand (Nameda®) is approved.
The text could be altered to:
“To date, there are five approved drugs available for AD, three of which are cholinesterase inhibitors, the fourth being an N-methyl-D-aspartate (NMDA) receptor antagonist and the fifth a combination of cholinesterase inhibitor and NMDA-receptor antagonist. “
- Page 2, second paragraph, line 3. “numerous clinical trials to identify DMT for AD have failed”. Should be altered to : “ numerous clinical trials to identify Disease Modifying Therapies (DMT) for AD have failed”.
- Page 3, second paragraph, line 8. “Carrying the APOE e4 variant significantly increases the lifetime risk for LOAD, whereas the number of e4 copies affects the level of risk and is associated with age of clinical disease onset [12,16], while APOE e2 conferred a protective effect [15-17].”
The authors mention APOE e2 only in this sentence. They neither mention nor discuss the possible therapeutic implications of the APOE e2 allele. Since the title of the paper is: “APOE; The new frontier in the Development of therapeutic Target Towards precision Medicine in LAOD”. I think that the review are lacking this aspect. It would be more complete as a review if the topic of APOE e2 as a potential therapeutic target was added.
For ref on this see:
- APOE2: protective mechanism and therapeutic implications for Alzheimer’s disease. Li Z, Shue F, Zhao N , Shinohara M and Bu G. Molecular Neurodegeneration (2020) 15:63 https://doi.org/10.1186/s13024-020-00413
- Serrano-Pozo A, Das S, Hyman BT. APOE and Alzheimer's disease: advances in genetics, pathophysiology, and therapeutic approaches. Lancet Neurol. 2021 Jan;20(1):68-80. doi: 10.1016/S1474-4422(20)30412-9. PMID: 33340485.
Author Response
Reviewer 1
I congratulate the authors for a paper that is very well written, interesting and well structured. I have a few, comments:
- Page 2, first paragraph, line 8. The authors state that: “there are five approved drugs available for AD, four of which are cholinesterase inhibitors” and “ the fifth being an N-methyl-D-aspartate (NMDA) receptor antagonist.”
To my knowledge, there are only 3 cholinesterase inhibitors approved by the FDA in the US (donepezil, rivastigmine and galantamine). In addition, 1 drug, combining memantine (NMDA-receptor antagonist) and donepezil, is available under the Brand Namazaric®. Also memantine (NMDA-receptor antagonist) in a single brand (Nameda®) is approved.
The text could be altered to:
“To date, there are five approved drugs available for AD, three of which are cholinesterase inhibitors, the fourth being an N-methyl-D-aspartate (NMDA) receptor antagonist and the fifth a combination of cholinesterase inhibitor and NMDA-receptor antagonist. “
OUR REPLY:
We altered the text according the reviewer suggestion. The revised text can be found in pages 1-2, and it reads:
“To date, there are only five FDA approved drugs available for AD, three of which are cholinesterase inhibitors (donepezil, rivastigmine and galantamine) [2], the fourth being an N-methyl-D-aspartate (NMDA) receptor antagonist [3], and the fifth a combination of cholinesterase inhibitor and NMDA-receptor antagonist. “
- Page 2, second paragraph, line 3. “numerous clinical trials to identify DMT for AD have failed”. Should be altered to : “ numerous clinical trials to identify Disease Modifying Therapies (DMT) for AD have failed”.
OUR REPLY:
We altered the text according the reviewer suggestion. The revised text can be found in page 2, and it reads:
“…numerous clinical trials to identify Disease Modifying Therapies (DMT) for AD have failed.”
- Page 3, second paragraph, line 8. “Carrying the APOE e4 variant significantly increases the lifetime risk for LOAD, whereas the number of e4 copies affects the level of risk and is associated with age of clinical disease onset [12,16], while APOE e2 conferred a protective effect [15-17].”
The authors mention APOE e2 only in this sentence. They neither mention nor discuss the possible therapeutic implications of the APOE e2 allele. Since the title of the paper is: “APOE; The new frontier in the Development of therapeutic Target Towards precision Medicine in LAOD”. I think that the review are lacking this aspect. It would be more complete as a review if the topic of APOE e2 as a potential therapeutic target was added.
For ref on this see:
- APOE2: protective mechanism and therapeutic implications for Alzheimer’s disease. Li Z, Shue F, Zhao N , Shinohara M and Bu G. Molecular Neurodegeneration (2020) 15:63 https://doi.org/10.1186/s13024-020-00413
- Serrano-Pozo A, Das S, Hyman BT. APOE and Alzheimer's disease: advances in genetics, pathophysiology, and therapeutic approaches. Lancet Neurol. 2021 Jan;20(1):68-80. doi: 10.1016/S1474-4422(20)30412-9. PMID: 33340485.
OUR REPLY:
We thank the reviewer for the great suggestion and in the revised manuscript we included a paragraph that discusses the aspect of APOEe2 as a potential therapeutics strategy via introducing its protective effect. The new text can be found in page 4 and it reads:
“Noteworthy, APOE e2 has been identified as a longevity variant, associated with beneficial effects on cognition, and accumulating evidence suggested it protects against AD (reviewed in ). While the mechanisms driving its protective effect remain unclear, potential therapeutic strategies designed to leverage the protective effect of APOE e2 such as, viral-mediated overexpression of APOE e2 and gene-editing conversion of APOE e4 to e2, hold promise as treatment options for AD. With that said, APOE e2 increases risk of other diseases including neurological disorders, thus, long-term safety concerns should be carefully evaluated when considering treatments inspired by the protective role of e2 in AD. This topic has been extensively reviewed elsewhere and is outside the scope of this review.
In the current review we focus on therapeutics strategies designed to mitigate the risk of LOAD conferred by the APOE gene (as proposed in Figure 2A). Below we describe three major technologies aim at targeting APOE gene, transcript or protein, that may serve as proof-of-concept for prospective therapeutics approaches (Figure 2B).”
Reviewer 2 Report
This is a timely review on treatment strategies targeting APOE for Alzheimer's disease. The review is well presented, covers recent work and is pleasant to read.
Some recommendations:
- While APOE is well known to the people most likely to read this review, I would recommend briefly recapitulating the physiological functions of APOE in section 2.1, particularly for readers to appreciate whether therapeutically reducing APOE could have any unwanted effects. Moreover, this balance between the advantages and disadvantages of targeting APOE expression would also be an interesting addition to the Discussion section.
- Similarly, in section 2.1 it would be helpful to mention the amino acid changes determining the APOE e2/e3/e4 alleles and their impact on protein structure, and to clarify this in the legend of figure 2a as well.
- For the general sections on ASOs and mAbs, shortcomings are briefly mentioned as 'robust knockdown and significant adverse side effects'. Please spend a few sentences explaining the problem of robust knockdown and illustrating which adverse side effects have been reported.
- For the section on gene editing, no shortcomings are mentioned explicitly. Please add a short elaboration on that.
- In the conclusion the authors mention that the ability to determine APOE alleles will enable to define people who will be responsive to APOE therapeutic approaches. However, they also describe that reduction of APOE expression has effects regardless of APOE genotype. It would be interesting to speculate how to determine who will benefit from such approach, especially in preclinical phases. Any progress on APOE as a biomarker?
- The illustrations in the paper enhance the review, but the font size is sometimes quite small, particularly in Figure 2. Further on Figure 2A, I recommend increasing the thickness of the arrows when indicating higher expression of e3, and/or show more than 1 e3 isoforms to indicate increased expression.
- Page 7/16, first line: "...the development of ASP..." - do the authors mean ASO?
Author Response
Reviewer 2
This is a timely review on treatment strategies targeting APOE for Alzheimer's disease. The review is well presented, covers recent work and is pleasant to read.
Some recommendations:
- While APOE is well known to the people most likely to read this review, I would recommend briefly recapitulating the physiological functions of APOE in section 2.1, particularly for readers to appreciate whether therapeutically reducing APOE could have any unwanted effects. Moreover, this balance between the advantages and disadvantages of targeting APOE expression would also be an interesting addition to the Discussion section.
OUR REPLY:
We agree with this constructive suggestion and accordingly included a new introductory paragraph to section 2, new subhead 2.1., that includes a brief description of APOE functions.
The new text can be found in page 3 and reads:
2.1. ApoE protein: function and isoforms
The apolipoprotein E protein (ApoE) has multiple functions and plays key roles in lipid metabolism, neurobiology, and neurodegenerative diseases. Its major function is to transport lipids among various cells and tissues of the body. In addition, intracellular ApoE may modulate various cellular processes physiologically or pathophysiologically, including cytoskeletal assembly and stability, mitochondrial integrity and function, and dendritic morphology and function. Overall, ApoE is widely involved in human health and disease.
- Similarly, in section 2.1 it would be helpful to mention the amino acid changes determining the APOE e2/e3/e4 alleles and their impact on protein structure, and to clarify this in the legend of figure 2a as well.
OUR REPLY:
Thank you for the useful advice. The new subhead 2.1., provides background information regarding the amino acid differences that define the three APOE isoforms.
The new text can be found in page 3 and reads:
“ApoE is encoded by the APOE gene positioned on chromosome 19q13.32 (GRCh 38: chr19:44,905,795-44,909,392). Two common coding SNPs in exon 4 of the gene give rise to three allelic variants, APOEe2, APOEe3, and APOEe4, encoding three corresponding protein isoforms that differ at two amino acid positions 112 and 158: ApoE2 (Cys112; Cys158), ApoE3 (Cys112; Arg158), and ApoE4 (Arg112; Arg158). It was suggested that the single amino acid change (position 112) between ApoE3 to ApoE4 protein isoforms, resulted in structural differences involve the interactions between sequences from both the N- and C-terminal domains (Figure 2A). “
According to the reviewer’s suggestion we also amended the legend of Fig 2A as follow:
“of APOEe4 and APOEe3 (differ in amino at acid position 112 Arg and Cys, respectively)”
- For the general sections on ASOs and mAbs, shortcomings are briefly mentioned as 'robust knockdown and significant adverse side effects'. Please spend a few sentences explaining the problem of robust knockdown and illustrating which adverse side effects have been reported.
OUR REPLY:
This is a thoughtful comment.
Since ApoE has a normal physiological function we anticipate that robust knockdown that eliminate the production of the protein will be problematic. We rearticulate this sentence to make a careful statement.
The new text can be found in page 5 and it reads:
“…, foremost, effective ASO delivery methods for the CNS remain a challenge as ASOs lack the ability to penetrate the blood-born barrier (BBB) efficiently [60]. Other concern is the possibility of adverse side effects due to ASOs induced-cellular toxicities and off-target effects in both sequence- and chemistry-dependent manners [60]. In addition, the potential robust knockdown mediated by ASOs can be deleterious resulting in deficiency in normal physiological levels of the target protein that is needed to maintain normal biological processes and cellular function. For example, RNAi studies reported neurotoxicity associated with robust reduction of SNCA levels65, 66 and suggested the need to maintain normal physiological expression levels of SNCA protein.”
However, this shortcoming is less relevant for the mAb treatment that target specifically protein aggregates. Thus, we removed this text from the discussion of shortcomings related to the mAb approach.
- For the section on gene editing, no shortcomings are mentioned explicitly. Please add a short elaboration on that.
OUR REPLY:
We thank the reviewer for this important suggestion. We included new paragraph that describes explicitly shortcoming of genome editing in clinical applications.
The pertained new text can be found in page 9:
“While the breakthrough technology of genome editing brings profound therapeutics opportunities to treat, cure and prevent genetic diseases, there are challenges that affect the translation into clinical applications. At present, major hindering aspects of therapeutics genome editing include accuracy (off target events), precision (undesired genomic sequence change), safety (e.g., immunogenicity) efficacy, efficient delivery systems, and extreme costs. The opportunities and challenges have been recently reviewed in detail elsewhere. “
- In the conclusion the authors mention that the ability to determine APOE alleles will enable to define people who will be responsive to APOE therapeutic approaches. However, they also describe that reduction of APOE expression has effects regardless of APOE genotype. It would be interesting to speculate how to determine who will benefit from such approach, especially in preclinical phases. Any progress on APOE as a biomarker?
OUR REPLY:
This is a valuable point.
Patients that will benefit from these strategies are carrier of APOE e4. As discussed in the concluding remarks.
The pertained text can be found in page 11:
“The ability to precisely characterize the APOE genotypes and determine carriers of the e4 risk allele facilitates the identification of the patients’ group that suffers from LOAD due to APOE and hence will be potentially responsive to treatment regimen that targets APOE.”
However, although there are evidences that link increased APOE expression with LOAD risk (discussed section 2.3 pages 3-4), currently there are still no ApoE levels-based biomarker to stratify individual at high risk due to APOE overexpression.
- The illustrations in the paper enhance the review, but the font size is sometimes quite small, particularly in Figure 2. Further on Figure 2A, I recommend increasing the thickness of the arrows when indicating higher expression of e3, and/or show more than 1 e3 isoforms to indicate increased expression.
OUR REPLY:
We made changes to Figure 2 to address the reviewer comment. Specifically, we increased the font size and refined the illustrations.
- Page 7/16, first line: "...the development of ASP..." - do the authors mean ASO?
OUR REPLY:
We thank the reviewer for identifying this typo. We made the correction to read ASO.